# Evaluating Influence of Inverter-based Resources on System Strength Considering Inverter Interaction Level

**Dohyuk Kim, Hwanhee Cho , Bohyun Park and Byongjun Lee \***

School of Electrical Engineering, Anam Campus, Korea University, 145 Anam-ro, Seongbuk-gu, Seoul 02841, Korea; thehyuk@korea.ac.kr (D.K.); whee88@korea.ac.kr (H.C.); wind833@korea.ac.kr (B.P.)

\* Correspondence: leeb@korea.ac.kr; Tel.: +82-10-9245-3242

**Abstract:** The penetration of renewable energy sources (RESs) equipped with inverter-based control systems such as wind and solar plants are increasing. Therefore, the speed of the voltage controllers associated with inverter-based resources (IBRs) has a substantial impact on the stability of the interconnected grid. System strength evaluation is one of the important concerns in the integration of IBRs, and this strength is often evaluated in terms of the short circuit ratio (SCR) index. When IBRs are installed in an adjacent location, system strength can be weaker than evaluation by SCR. This study proposes an inverter interaction level short circuit ratio (IILSCR) method by tracing IBRs output flow. The IILSCR can accurately estimate system strength, wherein IBRs are connected in adjacent spots, by reflecting the interaction level between IBRs. The study also demonstrates the efficiency of IILSCR by applying this method to Institute of Electrical and Electronics Engineers (IEEE) 39 bus test system and future Korea power systems.

**Keywords:** inverter-based resources; inverter interaction level; renewable energy resources; system strength; voltage oscillation

## 1. Introduction

Power systems are undergoing rapid changes mainly due to the increase of inverter-based resources (IBRs) supply, such as wind and solar power generation. This increase in IBRs lowers the inertia and system strength, which in turn affects frequency and voltage stability; therefore, it is necessary to pay attention to IBRs concentration areas. In addition, since IBRs are affected by the environment, it is installed in windy areas with abundant sunlight. Therefore, it is located far from the load center and there are few synchronous generators in such areas. In other words, IBRs are installed in areas where the system strength is relatively weak. When connected to systems with low system strength, the output sensitivity of IBRs has a direct impact on the output of the point of interconnection (POI). Wind and solar power control loops with inverter-based control schemes provide reactive/active power injections that react almost instantly to voltage changes at the POI. This dynamic response can lead to high voltage sensitivity to reactive power in weak power systems; small changes in reactive power can cause large fluctuations in voltage. When multiple IBRs are connected to weak power systems, power system stability issues related to voltage stability and quality can be exposed to serious consequences. In Korea, The Ministry of Trade, Industry, and Energy (MOTIE) has announced the "New Renewable 3020 Plan," which aims to expand the proportion of renewable energy to 20% of the total generation by 2030 [1]. To achieve this goal, MOTIE plans to distribute 48.7 GW of renewable energy generation facilities from 2018 to 2030 and supply clean energy such as solar and wind power to more than 95% of renewable energy generation facilities. In order to accommodate

this large amount of renewable energy, it is necessary to evaluate the system strength beforehand and prepare measures to cope with any problems. In particular, large-capacity wind and solar generators are concentrated in certain areas and are likely to interact with renewable energy sources (RESs) in the vicinity. Therefore, when IBRs are concentrated in electrically connected areas, evaluation of system strength becomes necessary to reflect the interaction of IBRs. In the meantime, many studies have been undertaken to assess system strength as a means of accessing renewable energy and identify potential problems. Research has been focused on the case of output and voltage oscillation by renewable generators, specifically when connected to weak systems [2–4]. IEEE [5] first introduced the concept of short circuit ratio (SCR) to evaluate Active Current / Direct Current (AC/DC) system strength when IBR is connected to the grid. However, SCR do not reflect the interaction impact of IBRs in the vicinity. In the case of an integrated system based on inverters, in [6], Saad et al. evaluated system strength using the interaction factor. Researchers [7–9] overcame the disadvantage of solely considering IBR capacity connected to the bus, i.e., the SCR approach. Methods such as weighted short circuit ratio (WSCR) were developed by Electric Reliability Council of Texas (ERCOT), taking into account the interaction between IBRs. The study [10,11] considered the actual electrical interaction when IBRs are connected to nearby areas. Impedance metrics were used to estimate the interaction from IBRs installed in the vicinity and evaluate weak systems. Several researchers [12–15] have presented extensive research on the voltage stability issues that occur when large wind farms are connected to weak grids. The above methodology was mainly a measure of the system strength for renewable energy connections. Several studies [16–19] have proposed a system strength evaluation and design for linking high-voltage direct current (HVDC) and wind farms. Certain studies [20–22] have introduced inverter parameters and controls that affect the stability of inverter-based equipment connected to the grid.

The existing methods for evaluating the system strength does not reflect the interaction of nearby IBRs, or assumed IBRs to have a 100% interaction within a boundary. However, it is very difficult to calculate the boundary within the actual system, and different results will be derived depending on the range of boundary. Therefore, to overcome these challenges, it is necessary to calculate the exact effect from nearby IBRs. In this paper, we propose a method to calculate the interaction level by tracking the output of IBRs. Power tracing method was used to reflect the impact of nearby IBRs. Thus, the interaction level of the IBRs can more accurately estimate the system strength when the renewable generator is connected to adjacent points. In addition, this study establishes a future system based on the power system of Southwest, the region where Korea's renewable energy will be concentrated. Modeling of the connected IBRs adopts Western Electricity Coordinating Council (WECC) Type 4 to comply with grid codes. In this system, the proposed method to evaluate system strength has been derived. Thus, it can be quantitatively analyzed for system voltage stability. Based on the analysis results, the accuracy of the system strength evaluation limits the number of renewable connections at the POI. Therefore, the risk of the renewable energy at the POI can be minimized when planning the grid. Finally, the dynamic simulation results verify the performance of the proposed method. The rest of this paper is organized as follows. Section 2 formulates a new methodology to overcome existing methodology limitations. Section 3 evaluates and verifies the system strength in IEEE 39 bus test system and South West region of Korea. Section 4 discusses the causes of reactive power oscillation in IBRs. Finally, this paper concludes with a brief explanation in Section 5.

## 2. Techniques Pertaining to the Interactions among the Inverter-Based Resources

### 2.1. Relationship Between Inverter Interaction Level and System Strength

This study aims to accurately evaluate the system strength when IBRs are connected to the system. When renewable power plants are installed in nearby areas, and if the system strength cannot be

accurately identified, it may cause voltage problems such as control stability and control interaction. The general system strength evaluation method is based on the following formula.

$$\text{SCR}_i = \frac{SCMVA_i}{P_{IBR_i}} \tag{1}$$

where $SCMVA_i$ is the short circuit capacity at the POI without the current contribution of the inverter-based resource, and $P_{IBR_i}$ is the nominal power rating of the IBR being connected at the POI. This system strength evaluation method is very useful when one IBR is connected to the system as shown in Figure 1. When evaluating the system strength through SCR, only the connection capacity at the POI is considered. As shown in Figure 2, it is easy to determine whether the system strength is strong through the PV curve. However, because this method does not reflect the interaction between the IBRs installed nearby, it may not provide an accurate evaluation of the system strength. In order to solve this problem, ERCOT has developed a WSCR method to evaluate the system strength reflecting the interaction between the wind farms in the Panhandle region.

$$\text{WSCR} = \frac{\sum_i^N SCMVA_i * P_{IBR_i}}{\left(\sum_i^N P_{IBR_i}\right)^2} \tag{2}$$

where $SCMVA_i$ is the short circuit MVA at bus $i$ before the connection of IBR and $P_{RES_i}$ is the MW rating of $IBR_i$ to be connected. $N$ is the number of IBRs fully interacting with each other and $i$ is the IBR index. The proposed WSCR calculation method is based on the assumption of full interaction between the IBRs as shown in Figure 3. This is equivalent to assuming all IBRs are connected at the same POI. The SCR obtained with this method provides a more conservative estimate of the system strength. For a real power system, there is typically some electrical distance between POIs and all IBRs will not fully interact with each other. However, the other jurisdiction of ERCOT, South Texas, is not considered because application of this method is difficult in this area. It is also becoming increasingly difficult to apply this method in the Panhandle region due to the growing number of transmission lines, which is obscuring the boundaries required for calculating WSCR. Thus, the existing method has two limitations, which are as follows:

- SCR, the basic method, can evaluate the system strength through a simple method; however, it is difficult to apply due to the large-scale, large-capacity, and power electronics-based facilities of the renewable generators that cause interaction effects.
- WSCR is the method wherein the connection buses are equalized and weighted to reflect the fully interaction effects of renewable generators installed in the vicinity. However, it is difficult to clearly calculate for the boundaries being equalized. Moreover, the result may be very different depending on the boundary setting.

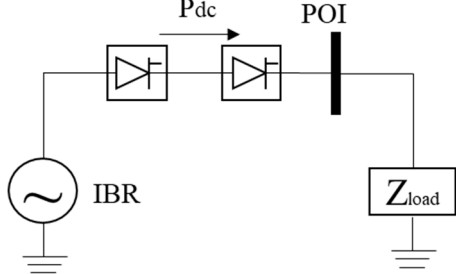

**Figure 1.** Simple AC System with single inverter-based resources (IBR).

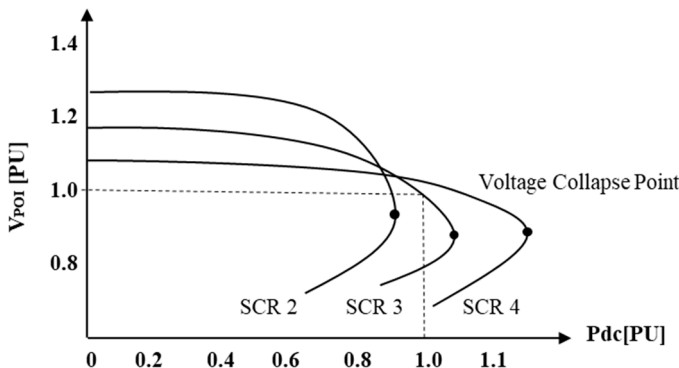

**Figure 2.** Voltage (point of interconnection (POI)) versus DC power curve.

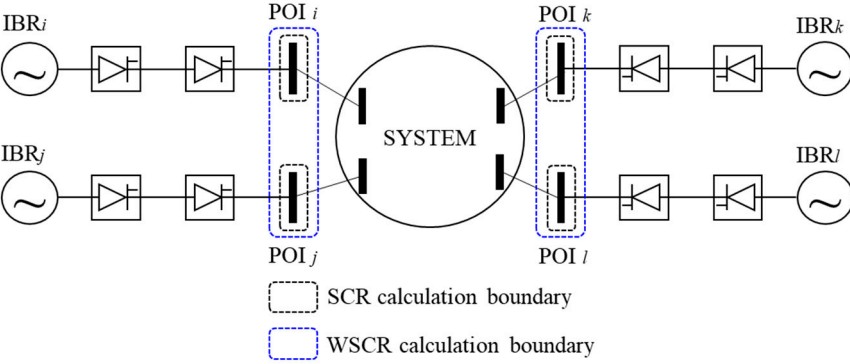

**Figure 3.** General system strength evaluation method for simple AC System with multiple IBRs.

The IBRs such as wind and solar plants are connected to power systems through power electronic controllers. The voltage/reactive power control loops within these IBRs are capable of providing almost instantaneous reactive power injections in response to the voltage change at the POI. Such rapid dynamic responses could result in high voltage sensitivity with respect to reactive power, i.e., a small change in reactive power will lead to a large oscillation in voltage. Therefore, the speed of the voltage controllers associated with IBR has a substantial impact on the stability of the interconnected grid. When a large amount of IBRs are connected to the weak points of a power system, undesirable system stability issues, especially those related to voltage stability and quality may be exposed and this may result in serious consequences. In the worst-case scenario, such system stability issues could be wide-spread and lead to loss of power generation and/or damage of IBR equipment if adequate protective measures are not implemented. Placing renewables at a distance from the main grid can result in problems when controlling the power injected to the grid. Moreover, the POIs of these IBRs are usually weak, hence voltage stability issues are more likely to occur in case of oscillation of reactive power. In the planning stage, therefore, it is important to evaluate the system strength that reflects the interaction of IBRs.

### 2.2. Techniques to Analyze the Inverter Interaction Level

Line power flow tracing algorithm is a useful method to determine the distributive path of the nodes from a specific generator to the final consumption of the electric load. Using this algorithm, it is possible to gain information about an IBR's effect on other buses. Equation (3) shows the amount of MW inflow from another IBR. The detailed derivation of (3) is presented in the Appendix A. Therefore, it is

feasible to decompose the effect of only one IBR from those of other components. Figure 4 describes the mechanism of interaction between IBRs at the power systems.

$$
\begin{aligned}
\lceil P_{m-i} \rceil &= \frac{\lceil P_{m-i} \rceil}{P_i} P_i = \frac{\lceil P_{m-i} \rceil}{P_i} \sum_{m=1}^{N} \lceil A_u^{-1} \rceil_{im} P_{G_m} \\
&= \alpha_{i1} P_{G1} + \alpha_{i2} P_{G3} + \cdots\cdots + \alpha_{im} P_{Gm} \\
&where\ 0 \cdots \alpha_{im} \cdots 1\ for\ m = 1, 2, \cdots\cdots, N
\end{aligned}
\tag{3}
$$

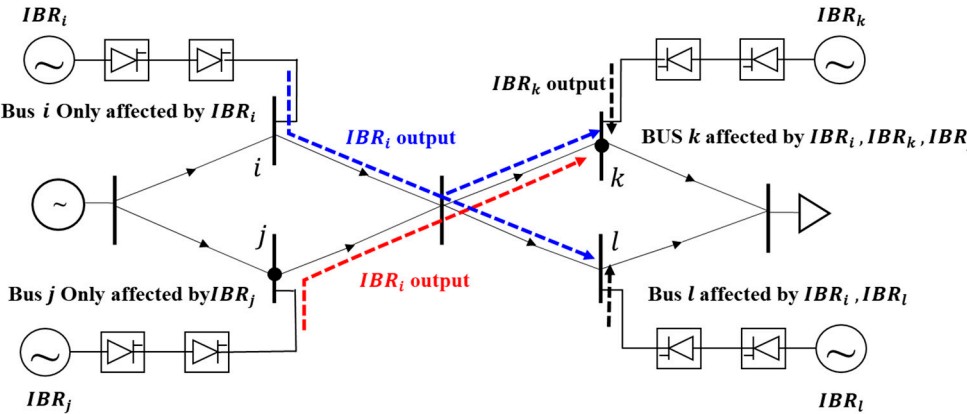

**Figure 4.** Concept of inverter-based resource interaction.

## 2.3. Inverter Interaction Level Short Circuit Ratio

Interaction level short circuit ratio (IILSCR) tracks the amount of power output from the IBRs to reflect the interactions between the IBRs. The output of the IBR is divided based on a matrix that reflects the system's lines and loads, allowing for accurate mutual impacts. Therefore, as shown in Figure 5, it is possible to reflect the output of IBRs flowing from nearby as well as the amount of renewable connections connected to the bus. Therefore, equalization of renewable energy resources installed in the vicinity and calculation of the boundaries are not required. Equation (4) shows a new system strength method that reflects the interaction between IBRs.

$$
\text{IILSCR}_i = \frac{SCMVA_i}{P_{IBR_i} + \sum_{m=1, m \neq i}^{N} P_{IBR_{m-i}}}
\tag{4}
$$

where $SCMVA_i$ is the short-circuit capacity connected to IBR on bus $i$, $P_{IBR_i}$ is the capacity of the IBR installed on bus $i$, and $P_{IBR_{m-i}}$ is the inflow from the nearby IBR. In the power system, the system is generally considered weak if the SCR is less than 3 and weak if the WSCR is less than 1.5. Since IILSCR does not equalize buses like WSCR, the system strength can be determined by individual bus evaluation method such as SCR as shown in Figure 6.

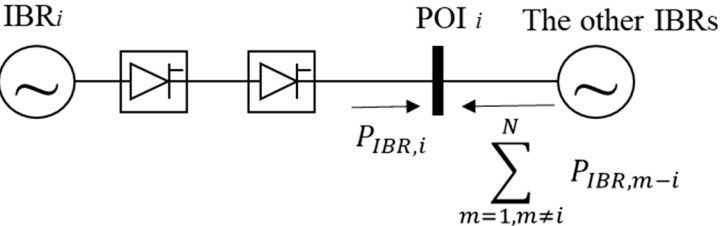

**Figure 5.** Equivalent diagram at POI considering between multiple IBRs.

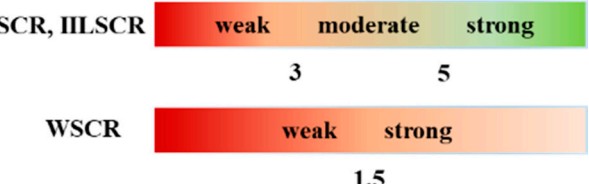

**Figure 6.** Weak system criteria for various indexes.

## 3. Numerical Studies

In this section, the proposed method is validated. The wind and solar generator to be installed is a WECC Type 4 (fully rated converter) generator model. According to the Korean grid code [23], the power factor is modeled as a power factor of ± 0.95 for wind generators and as 1 for solar generators. The detailed Korean grid code about reactive power requirements are presented in Appendix B. Applied to the IEEE 39 bus test system and the Korean power system, vulnerable buses were selected in each system by comparing SCR, WSCR, and IILSCR.

### 3.1. Analysis of the IEEE 39 Bus Test System

In order to verify the effectiveness of IILSCR, we applied it to the IEEE 39 bus test system [24] as shown in the Figure 7. Conventional generators installed at buses 35 and 36 have been replaced by WECC Type 4 generator model. As shown in Table 1, the IBR output was limited so that the SCR value of the buses to be connected was 5, 4, 3. Bus 36 is not affected by IBR output connected at bus 35. However, bus 35 is affected by IBR output connected at bus 36. All three cases are considered strong in SCR (above 3) and WSCR (above 1.5), but in case of IILSCR, risk is indicated in CASE3.

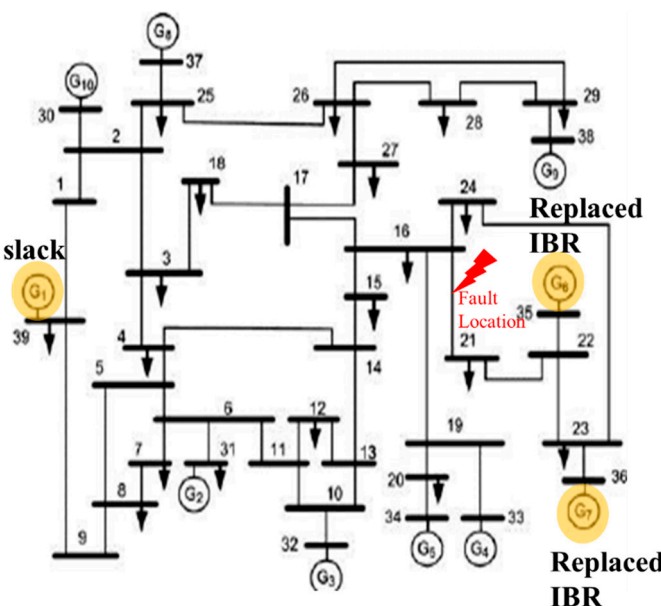

**Figure 7.** A Single-line diagram of IEEE 39Bus Test System.

Figures 8–10 show active power output, reactive power output, and voltage magnitude in detailed time-domain dynamic simulation following a fault in system. As can be seen in the case of Case 2, IILSCR 3.33, it recovers stably after fault. However, in the case of CASE3 where IILSCR becomes 2.24, it can be seen that oscillation occurs after fault.

**Table 1.** Comparison among various indices for IEEE 39 bus test system.

| Year | Bus No. | SCC (MVA) | IBR Capacity (MW) | SCR | WSCR | Inflow from Nearby IBRs (MW) | Total IBRs Inflow (MW) | IILSCR |
|------|---------|-----------|-------------------|-----|------|------------------------------|------------------------|--------|
| CASE1 | 35 | 2127 | 452.61 | 5 | 2.2 | - | 452.61 | 5 |
|       | 36 | 1624 | 382.76 | 5 |     | 86.26 | 469.02 | 3.95 |
| CASE2 | 35 | 2127 | 531.64 | 4 | 2.0 | - | 531.64 | 4 |
|       | 36 | 1624 | 405.93 | 4 |     | 81.51 | 487.44 | 3.33 |
| CASE3 | 35 | 2127 | 708.85 | 3 | 1.6 | - | 708.85 | 3 |
|       | 36 | 1624 | 541.24 | 3 |     | 73.66 | 614.90 | 2.64 |

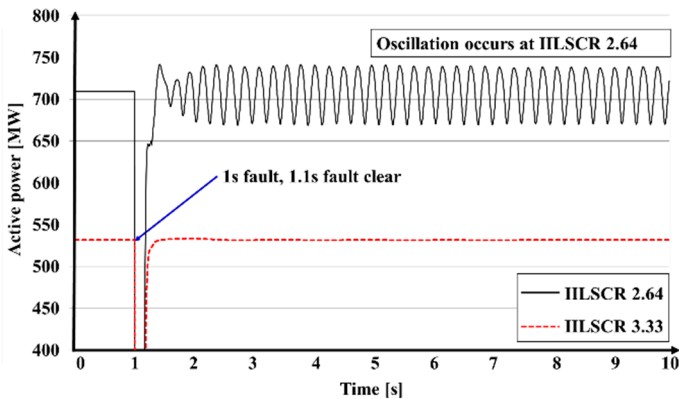

**Figure 8.** Active power response at bus 36 after fault for IEEE 39 bus test system.

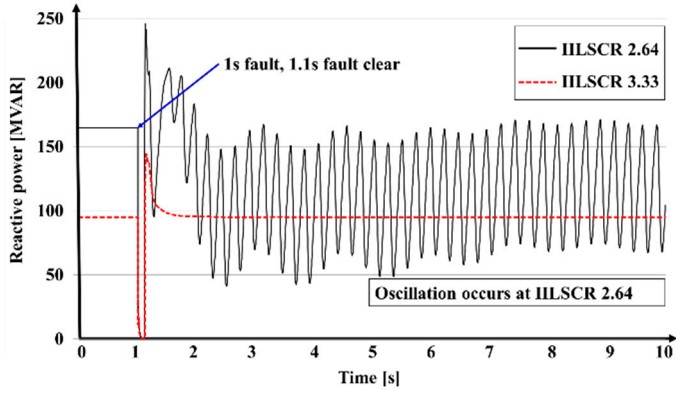

**Figure 9.** Reactive power response at bus 36 after fault for IEEE 39 bus test system.

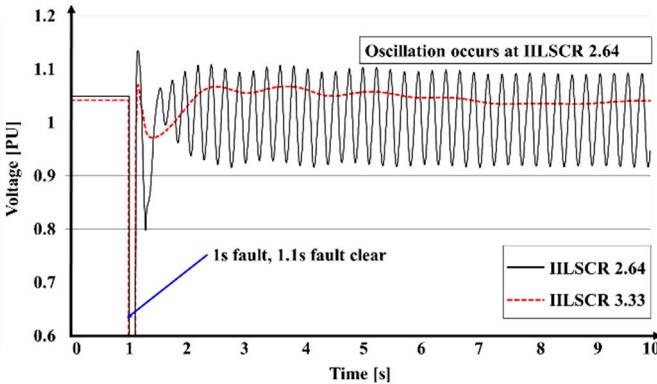

**Figure 10.** Voltage response at bus 2 after fault for IEEE 39 bus test system.

### 3.2. Analysis of the Korea Power System

Korea's MOTIE has announced the 'New Renewable 3020 Plan' to increase the share of renewable energy to 20% of total generation by 2030. To this end, MOTIE plans to supply 48.7 GW of renewable energy generation facilities. More than 95% of the renewable energy power generation facilities, which will be installed from 2018 to 2030, will be sourced by clean energy such as solar and wind power. The southwest region, in which most of the installation is expected, is the region farthest from the load center and without any synchronous generators. Figure 11 shows the southwest regional system one-line diagram (345 kV, 154 kV) in Korea and shows the highest concentration of IBRs in the southwest. To verify the effectiveness of IILSCR, the system strength of IBRs, which will be installed in the southwest region, is evaluated. Figure 12 is a schematic of the Southwest region where the renewable energy resources are installed. The buses 1 to 10 were selected in study areas. Table 2 and Figure 13 show the results of the system strength evaluation based on the amount of renewable power generation to be installed in the study area. The Short Circuit Capacity (SCC) value, which represents the strength of the AC system, gradually increases over the years; this is the result of the system being reinforced gradually. However, the system reinforcement does not occur, and the connection amount of IBRs is observed to increase. From 2019 to 2021, SCR, WSCR, and IILSCR were all higher than the weak criteria. However, when comparing the results of 2022, SCR and WSCR were evaluated as system strength at bus 2, while IILSCR was less than 3. The different results regarding IILSCR from the previous two methods can be attributed to the difference between how these circuit ratios reflect interactions. In the case of the SCR, it only reflected the amount connected to its bus, which resulted in relatively positive results. The second cause is the effect of the boundaries of the buses being equalized in the WSCR. In this study, the WSCR calculation is equalized only when a renewable generator is connected within one level away from the bus. If equalized within three levels, the results would be different. In the case of IILSCR, equalization is not required; hence, the system strength on each bus can be assessed and consistent results obtained.

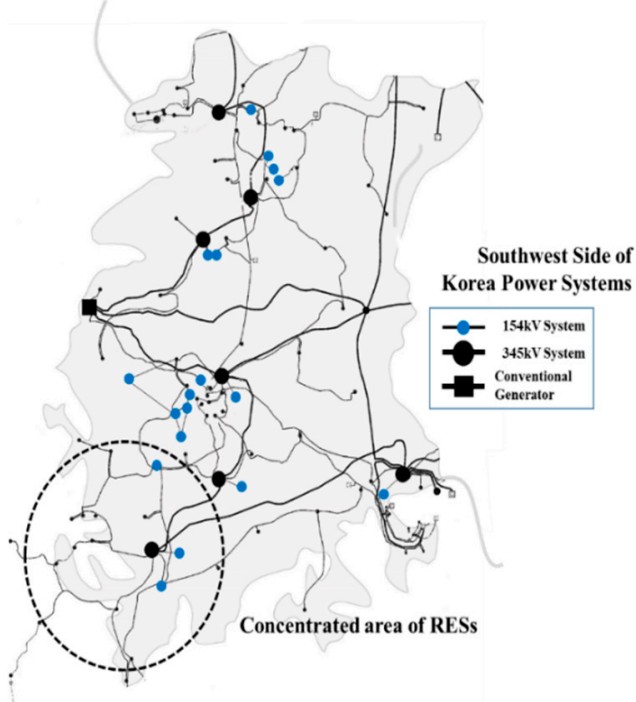

**Figure 11.** The configuration of southwest side of Korea power systems transmission grid.

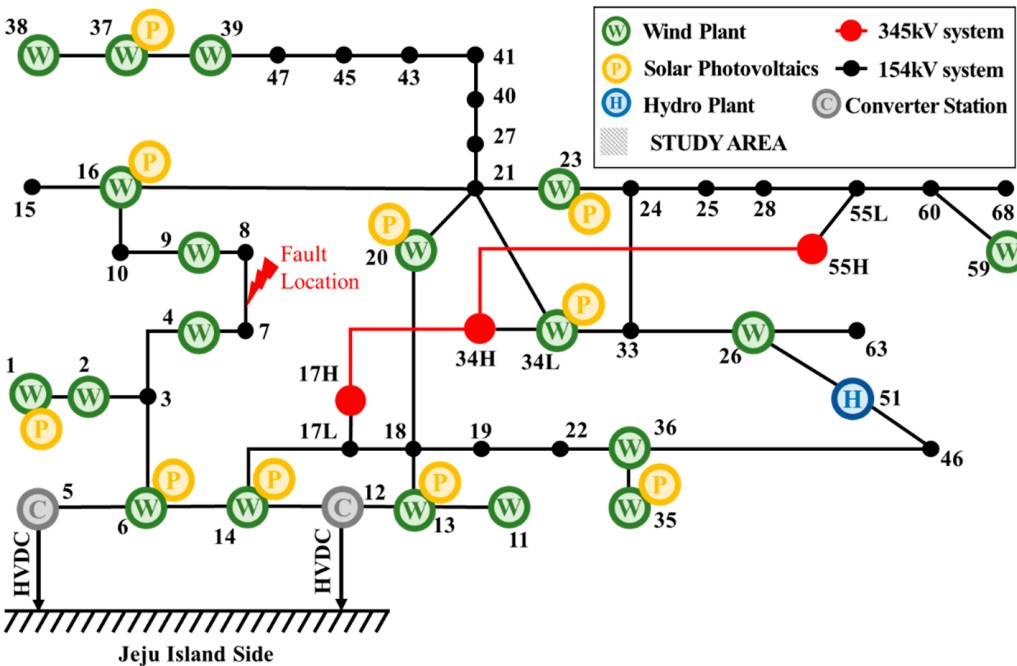

**Figure 12.** A Single-line diagram of the studied area for future renewable energy source (RES) connection.

**Table 2.** Comparison among various indices for southwest side of Korea power systems.

| Year | Bus No. | SCC (MVA) | IBR Capacity (MW) | SCR | WSCR | Inflow from Nearby IBRs (MW) | Total IBRs Inflow (MW) | IILSCR |
|------|---------|-----------|-------------------|------|------|------------------------------|------------------------|--------|
| 2019 | 1 | 2090 | 150.0 | 13.93 | 10.81 | - | 150.0 | 13.93 |
|      | 2 | 2101 | 43.5 | 48.30 |       | 150.0 | 193.5 | 10.86 |
| 2020 | 1 | 2090 | 400.0 | 5.23 | 4.39 | - | 400.0 | 5.23 |
|      | 2 | 2101 | 76.0 | 27.65 |      | 400.0 | 476.0 | 4.41 |
| 2021 | 1 | 2094 | 400.0 | 5.24 | 4.40 | - | 400.0 | 5.24 |
|      | 2 | 2106 | 76.0 | 27.71 |      | 400.0 | 476.0 | 4.42 |
|      | 4 | 3576 | 60.0 | 59.60 | - | 239.3 | 299.3 | 11.95 |
|      | 6 | 2917 | 37.5 | 77.79 | - | 239.3 | 276.8 | 10.54 |
| 2022 | 1 | 2101 | 673.0 | 3.12 | 2.63 | - | 673.0 | 3.12 |
|      | 2 | 2112 | 268.0 | 7.88 |      | 673.0 | 941.0 | 2.24 |
|      | 4 | 3574 | 60.0 | 59.56 | - | 612.1 | 672.1 | 5.32 |
|      | 6 | 2920 | 37.5 | 77.88 | - | 289.5 | 327.0 | 8.93 |
|      | 9 | 2980 | 50.0 | 59.60 | - | 175.5 | 225.5 | 13.22 |
| 2023 | 1 | 2124 | 1112.5 | 1.91 | 1.78 | - | 1112.5 | 1.91 |
|      | 2 | 2136 | 268.0 | 7.97 |      | 1112.5 | 1380.5 | 1.54 |
|      | 4 | 3776 | 60.0 | 62.93 | - | 1009.8 | 1069.8 | 3.53 |
|      | 6 | 2984 | 37.5 | 79.57 | - | 371.2 | 408.7 | 7.30 |
|      | 9 | 3466 | 50.0 | 69.33 | - | 356.0 | 406.0 | 8.54 |

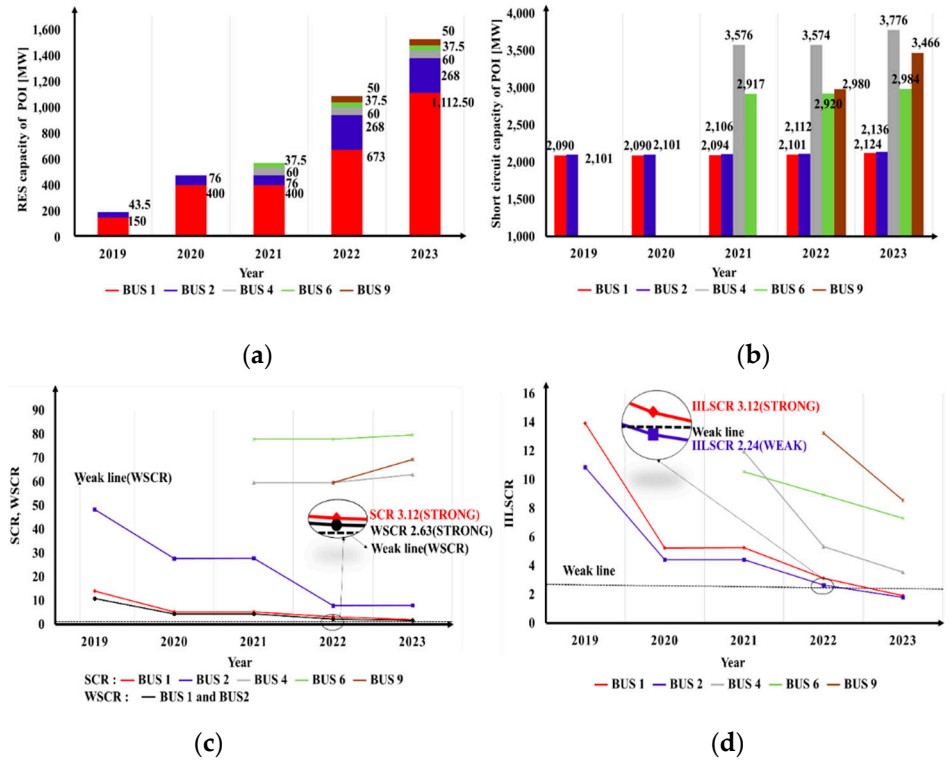

**Figure 13.** Annual values (IBR Capacity, SCC, short circuit ratio (SCR), weighted short circuit ratio (WSCR), and proposed interaction level short circuit ratio (IILSCR)) for the southwest side of future Korea power system (**a**) IBR capacity of point of interconnection (POI). (**b**) Short circuit capacity of POI. (**c**) Calculation of SCR and WSCR. (**d**) Calculation of the proposed IILSCR.

### 3.2.1. Validation through Dynamic Simulation

In the previous section, a dynamic simulation was explained to verify the differences in the system strength evaluation that occurred in 2022. Dynamic simulation assumes a single line 7–8 fault of 154 kV, three levels away from 2 buses in 1 s, and the fault was eliminated in 1.1 s. Bus 1 flows through bus 2 so that all the power generated by bus 1 can reach the center of load. As a result, bus 2 is affected by the amount of renewable energy generated by bus 1. Therefore, IILSCR of bus 2 is calculated by adding 673 MW of renewable power connected to bus 1 as well as 268 MW of bus 2. This would cause the IILSCR to decrease to 2.24, resulting in a very weak bus 2. In order for the IILSCR result to be greater than or equal to 3, it is necessary to limit the number of renewable connections on buses 1 and 2. In this study, we limit the number of IBRs connected to bus 1 because we propose a method for evaluating the system strength that reflects interaction. Therefore, bus 1 connection was limited from 673 MW to 400 MW. If the 273 MW connection is restricted, the IILSCR value for bus 2 will be 3.12. Figures 14–16 show dynamic simulation results for the IILSCR of bus 2 with 2.24 and 3.12. As shown in the figures, if the IILSCR is 3 or less, the original equilibrium point cannot be found after the failure and the voltage, reactive power, and active power are oscillated. Renewable generators based on inverters respond quickly to faults in the system using fast switching devices. If the renewable connection bus is not strong, the reactive power injected by the renewable generator can cause large changes, causing causes oscillations. However, when the IILSCR is 3 or higher, the influence of the renewable generators has less influence on the AC system. As shown in Figures 14–16, if IILSCR is 3 or more, after the fault, oscillation does not occur and a new equilibrium point is reached.

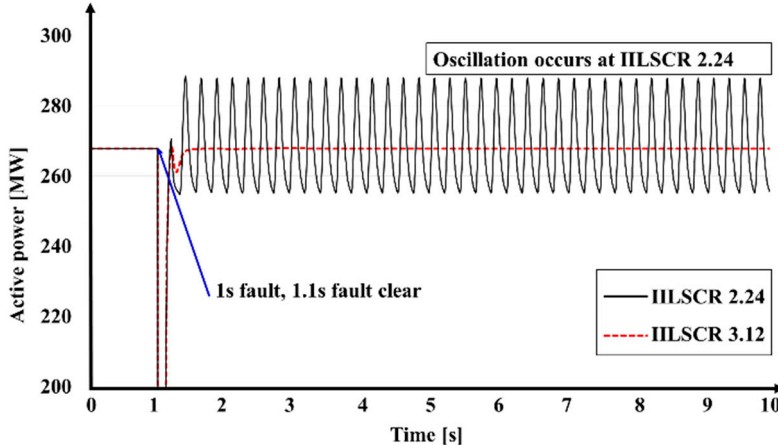

**Figure 14.** Active power response at bus 2 after fault for southwest side of Korea power systems.

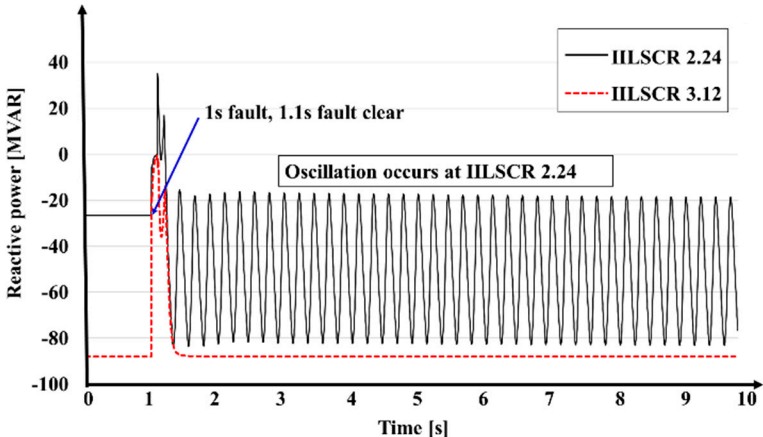

**Figure 15.** Reactive power response at bus 2 after fault for southwest side of Korea power systems.

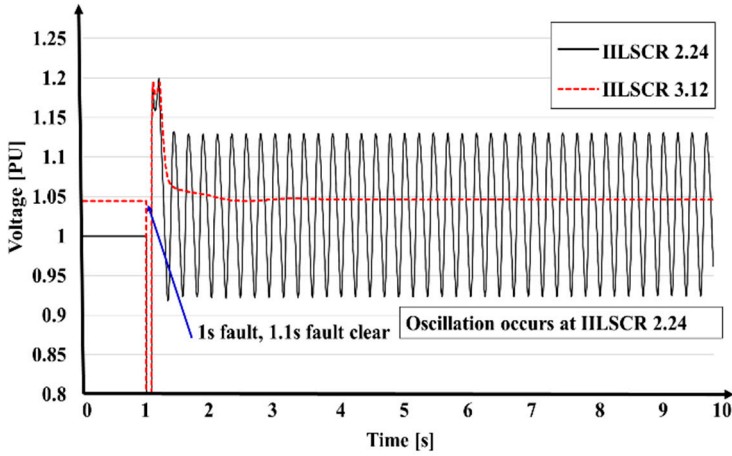

**Figure 16.** Voltage response at bus 2 after fault for southwest side of Korea power systems.

### 3.2.2. Oscillation Source Tracking

With dynamic simulation, if oscillation occurs on bus 2 after a fault, the oscillation will flow through the system. If oscillation occurs from the point of view of the system operator, the source of the oscillation should be identified to eliminate the cause and facilitate stable operation. Therefore, it is necessary to confirm whether the source of the generated oscillation, IILSCR, is from the lowest rated bus 2. To confirm this, the DEF (dissipating energy flow) method was applied. Using the DEF method

based on the energy approach, we can find sources of weakly damped natural or forced oscillations in the power system. After obtaining the values of voltage, frequency, reactive power, and active power in the bus through dynamic simulation, the source of oscillation was traced using the following equations [25,26]. From [25], an energy function form in the network has been derived as Equation (5) to express that the energy dissipates at the network by the damping torque.

$$W_{ij}^D = \int (\Delta P_{ij} d\Delta\theta_{ij} + \Delta Q_{ij} d(\Delta \ln V_i)) \tag{5}$$

where $\Delta P_{ij}$ and $\Delta Q_{ij}$ are deviations from the steady-state values of the active and reactive power flow in branch *i-j*; $\Delta\theta_i$ and $\Delta f_i$ are deviations from the steady-state values of bus voltage angle and frequency at bus *i*; $V_i$ is the bus voltage magnitude. $\Delta \ln V_i = \ln V_i - \ln V_{i,s}$, where $V_{i,s}$ is the steady-state voltage magnitude. Equation (6) is an approximated form of Equation (5) for the computational purpose [24].

$$\begin{aligned} W_{ij}^D &\approx \int\left(\Delta P_{ij} d\Delta\theta_{ij} + \Delta Q_{ij}\frac{d(\Delta V_i)}{V_i^*}\right) \\ &= \int\left(2\pi\Delta P_{ij}\Delta f_{ij} dt + \Delta Q_{ij}\frac{d(\Delta V_i)}{V_i^*}\right) \end{aligned} \tag{6}$$

From this stage, there are no need to take steady-state quantities, but a mean value for the studied frequency or period by applying fast Fourier transformation (FFT) does work sufficiently. From Equation (6), values with $\Delta$ means that the deviation between each of the mean values for studied period. Additionally, $V_i^* = \widetilde{V}_i + \Delta V_i$ and $\widetilde{V}_i$ is the average voltage in this studied period. Furthermore, Equation (6) is modified to consider discrete inputs and dissipating energy outputs for each step [26], by constructing a recurrence relationship as follows,

$$W_{i,j,k+1}^D = W_{i,j,k}^D + \left(2\pi\Delta P_{ij,k}\Delta f_{i,k} \cdot t_s + \Delta Q_{ij,k}\frac{\Delta V_{i,k+1} - \Delta V_{i,k-1}}{2V_{i,k}^*}\right) \tag{7}$$

where $t_s$ is the time step between samplings for the dynamic simulation and index *k* for each quantity reflects the time sample number for the time instant. Figure 17 shows the magnitude and direction of oscillation energy after failure. As with the IILSCR results, it can be seen that the oscillation energy is highest on bus 2. Figures 18 and 19 show the energy of increasing oscillations that occur after in the study area over time. Therefore, IILSCR shows that it is possible to evaluate the system strength more accurately when connecting renewable sources.

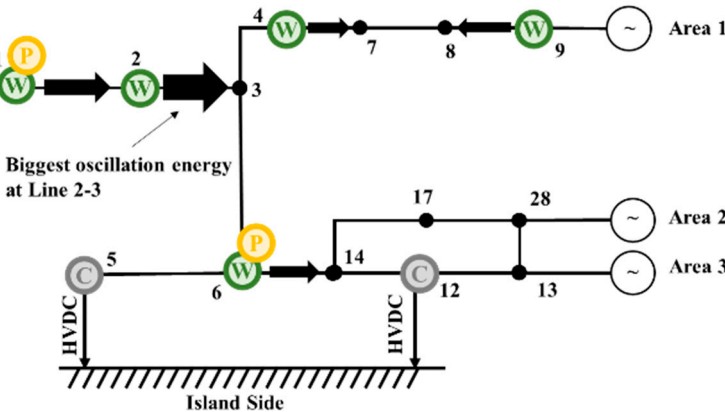

**Figure 17.** Oscillation energy flow.

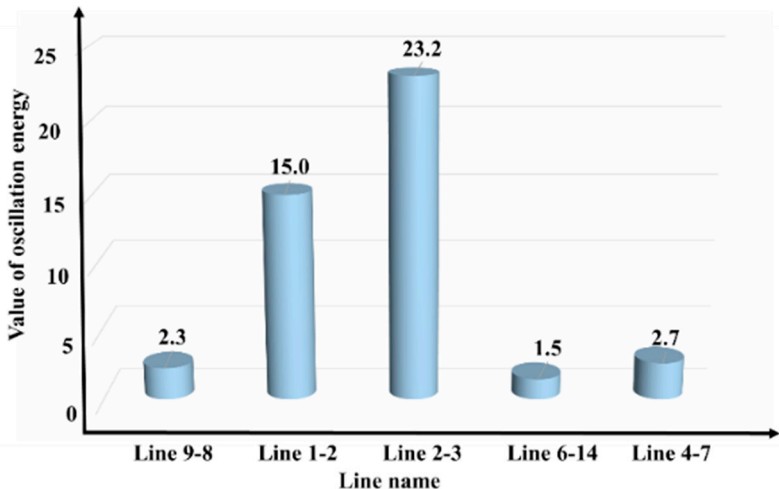

**Figure 18.** Last value of oscillation dissipating energy flow for 10 s of time interval.

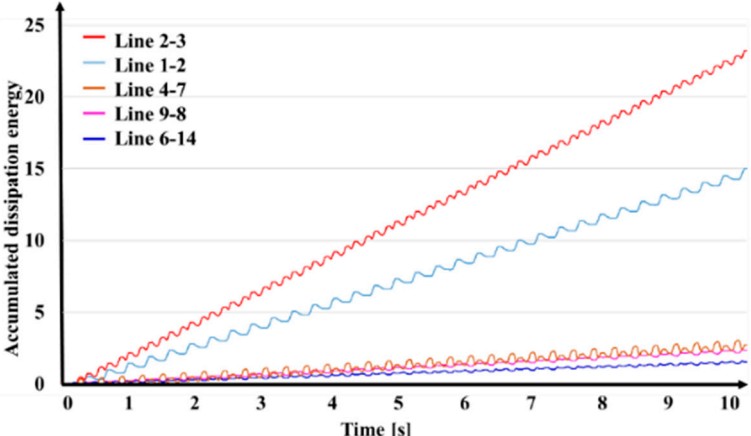

**Figure 19.** Calculation of oscillation dissipating energy after the fault.

## 4. Discussion

In the previous section, we introduced the proposed method to detect interaction phenomena including oscillation and verified the origin source. In this section, we discuss the relationship between voltage regulation deadband and the influence of interactions by changing deadband range. The Korean grid code for IBR contributes to voltage stability by supplying or absorbing reactive power when the POI voltage is out of the 0.95–1.05 pu range. Figure 20 shows the voltage sensitivity at the 2 bus in dynamic simulation. IBRs supply reactive power to recover the voltage in the transient period after a fault. When the system strength is low, the voltage sensitivity is increased, and oscillation is generated. This is because the reactive power absorption and supply of all nearby IBRs is beyond the deadband range. In order to prevent the oscillation, if the deadband is increased to 0.9 to 1.0 pu as shown in Figure 21, the oscillation may be eliminated. As mentioned earlier, increasing the deadband range may cause voltage oscillation problems, so it is necessary to accurately evaluate the system strength at the planning stage. To apply the proposed method to real power systems, the exact system topology is important.

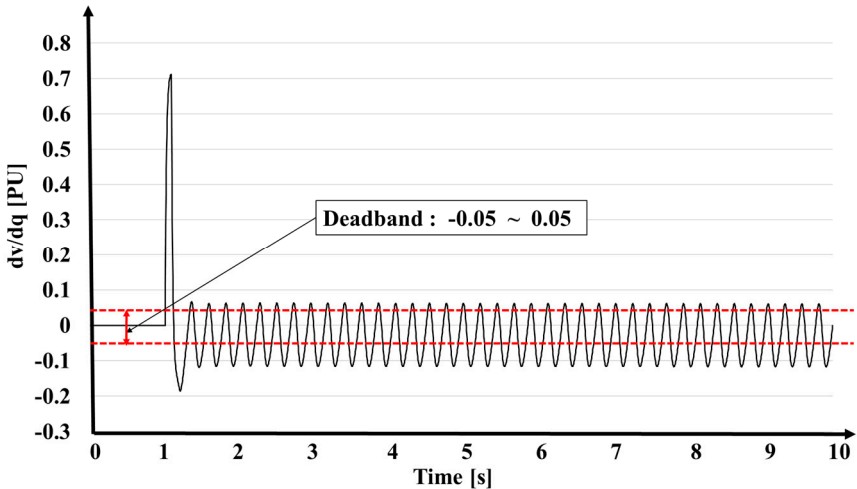

**Figure 20.** Voltage sensitivity at bus 2 during dynamic simulation for southwest side of Korea power systems.

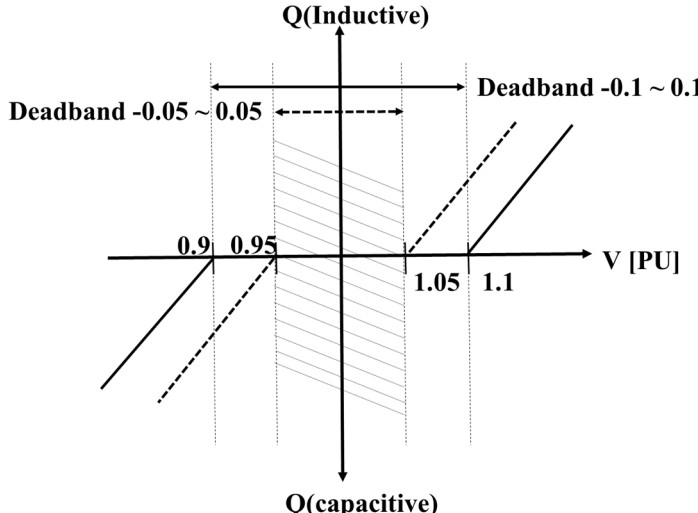

**Figure 21.** Reactive power voltage response capability for type-4 wind power plant.

## 5. Conclusions

In this study, a system strength evaluation method is derived that reflects the interaction effects between IBRs. This method was developed by tracing the output of IBR to accurately assess system strength. Subsequently, this method was applied to IEEE 39bus test system and the future Korea systems to compare the differences with the existing methodology. The energy decision method was used to analyze the oscillation source. As a result of simulations, the following conclusions can be drawn.

1.  The system strength evaluation by the proposed method is shown to correspond to the dynamic simulation results. It has been shown that if the IBRs are concentrated in areas with weak systems, oscillation problems representing voltage instability may occur.
2.  The energy dissipation method showed that the source of oscillation was consistent with the weakest bus of IILSCR. A bus can be the weakest bus since the bus is influenced by both the IBRs installed on its bus and nearby the bus.
3.  When the IBRs are concentrated in weak area, the deadband range can be selected to eliminate voltage instability such as voltage oscillation. However, this method can cause power quality problems. Therefore, the system strength must be accurately evaluated for areas in which

IBRs are concentrated. Therefore, the methodology proposed in this paper can serve as an adequate preliminary review to assess the system strength before adopting a detailed approach to system planning.

**Author Contributions:** D.K. conceived and designed the research methodology, performed the system simulations, and wrote this paper. B.L. supervised the research, improved the system simulation, and made suggestions regarding this research. H.C. and B.P. discussed the results and contributed to the writing of the paper. All authors have read and agreed to the published version of the manuscript.

**Funding:** This work was supported by Korea Electric Power Corporation [Grant no. R17XA05–4] and Human Resources Program in Energy Technology of the Korea Institute of Energy Technology Evaluation and Planning (KETEP), financed by the Ministry of Trade, Industry & Energy, Republic of Korea [Grant no. 20174030201540].

**Conflicts of Interest:** No conflicts of interest relevant to this article are reported.

## Appendix A

Derivation of (3)

The active power flow of bus $i$ is composed of the sum of MW generated from IBR at bus $i$ and MWs delivered from other IBR near bus $i$.

$$
\begin{aligned}
P_i &= \sum_{j\in\alpha_i^{(u)}} \lceil P_{i-j}\rceil + P_{G_i} \\
&= \sum_{j\in\alpha_i^{(u)}} \frac{\lceil P_{i-j}\rceil}{P_j}P_j + P_{G_i} \; for \; i = 1, 2, \cdots\cdots, n
\end{aligned}
\tag{A1}
$$

Affected by interaction at bus $i$ with bus $j$ can be expressed mathematically as follows [27,28]: The active power flows into the bus $i$ can be simply expressed as $P_{G_i}$. By multiplying the inverse matrix of distribution Au matrix and Pg vec0.tor, which describes the generation of each bus, gives $P$, the active power supplied from individual buses.

$$
P_{G_i} = P_i - \sum_{j\in\alpha_i^{(u)}} \frac{\lceil P_{i-j}\rceil}{P_j}P_j \;\Rightarrow\; P_G = A_u P.
\tag{A2}
$$

Au matrix is an $n \times n$ square upstream matrix for power distribution for all buses; individual elements are given as follows,

$$
\lceil A_u\rceil_{ij} = \begin{cases} 1 & for \; i = j \\ -\frac{\lceil P_{j-i}\rceil}{P_j} & for \; j \in \alpha_i^{(u)} \\ 0 & for \; otherwise \end{cases}
\tag{A3}
$$

Assuming that $A_u$ matrix is nonsingular, a vector representing the active power supplied at each bus P and the supplied MWs at each bus $P_i$ are shown as,

$$
P = A_u^{-1} P_G
\tag{A4}
$$

$$
P_i = \sum_{k=1}^{n} \lceil A_u^{-1}\rceil_{ik} P_{G_k}
\tag{A5}
$$

Additionally, the result of Equation (A6) is an inflow MWs at bus $i$, which decides how much of generation component is composed of that bus as a multiple of Au upstream matrix and generated

power. Using the approaches above, we can describe the inflow from the transmission line *i-j* to bus *i* by proportional sharing rule as,

$$
\begin{aligned}
\lceil P_{i-j} \rceil &= \frac{\lceil P_{i-j} \rceil}{P_i} P_i = \frac{\lceil P_{i-j} \rceil}{P_i} \sum_{k=1}^{n} \lceil A_u^{-1} \rceil_{ik} P_{G_k} \\
&= \alpha_{i1} P_{G1} + \alpha_{i2} P_{G3} + \cdots\cdots + \alpha_{in} P_{Gn} Z \\
&\quad where\ 0 \le \alpha_{ik} \le 1\ for\ i = 1, 2, \cdots\cdots, n
\end{aligned}
\tag{A6}
$$

Hence, the method is applied as a tracing method among the interaction buses with respect to the transmission lines. Moreover, Equation (A6) decides the contribution of *k*-th generator for the amount of MWs supplied to the transmission line *i-j*.

## Appendix B

Criteria for Reliability and Quality of Electricity System of Korea
Reactive power capability requirements:

- Wind generator: lag 0.95–lead 0.95,
- Tidal energy generator: lag 0.95–lead 0.95,
- Photovoltaic generator: none.

Low Voltage Ride Through standard is shown in Figure A1.

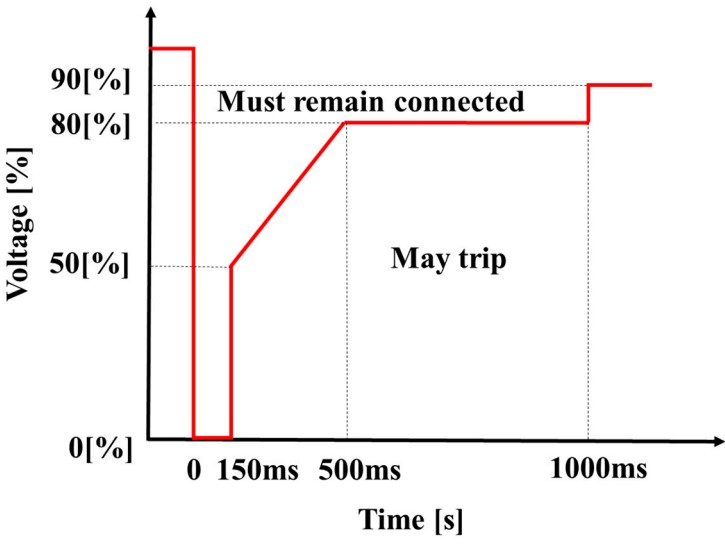

**Figure A1.** Low-voltage ride-through curve.

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
