# Peer review of "Evaluating Influence of Inverter-based Resources on System Strength Considering Inverter Interaction Level"

_sustainability, doi:10.3390/su12083469_

Round 1

Reviewer 1 Report

In this paper, a system strength evaluation method is derived that reflects the interaction effects between inverter-based resources (IBRs). The idea is interesting and useful but need some improvement considering the below points: 

  1. What are the limitations of previous research works? This section is missing in Introduction Section. 
  2. Describe the parameters of IEEE 39 Bus system. Atleast provide reference.
  3. The fault conditions are missing. 
  4. In numerical studies, the authors mentioned about grid code. What kind of grid is considered for PV system? 
  5. The dynamic simulation analysis should be verified through real wind speed and solar irradiance data.

Author Response

Point 1: What are the limitations of previous research works? This section is missing in Introduction Section.

 Response 1:

Thank you for your suggestion. We have revised the introduction part by adding the limitations of previous research works as follows: (page 2, Line 65-69)

Point 2: Describe the parameters of IEEE 39 Bus system. At least provide reference.

Response 2:

 We agree with your comment, so we have added reference. (page 6, Line 189 and page 18, Line 471-472)

Point 3: The fault conditions are missing.

Response 3:

 We agree with your comment, so we have changed the Figure 12 and added the sentences in the section 3.2.1 to improve the understanding. (page 8, Line 241-242 and page 9, Figure 12)

Point 4: In numerical studies, the authors mentioned about grid code. What kind of grid is considered for PV system?

Response 4:

 We agree with your comment about grid code. We have added a Korean grid code to the appendix to provide the basis for the power factor operating range for wind power and photovoltaic generator. (page 6, Line 183-187 and page 16-17, Appendix B and page 18, Line 469-470)

Point 5: The dynamic simulation analysis should be verified through real wind speed and solar irradiance data.

Response 5:

We agree with your comment about real wind speed and solar irradiance data. In this dynamic simulation, however, consistent wind speed and solar irradiance are assumed to represent rated power output. The reason for this is the higher the power output of the renewable energy source, the lower the system strength as follow equation. With less wind and irradiance, the renewable energy source cannot generate the power and have no effect to the system. Therefore, this dynamic simulation focuses on rated power output. Because we can get a conservative results considering the worst case. The content proposed by the reviewer will be advanced in future research.

 We have attempted to correct all of the errors throughout the text. We hope the revised manuscript will better meet the requirements of your journal for publication. We thank the editor and the reviewer of the Sustainability once again for the constructive review of our paper.

 Sincerely yours,

Dohyuk Kim, PhD candidate

Department of Engineering

Korea University

Seoul, Korea

Tel : 82-2-3290-3697

E-mail : [email protected]

Reviewer 2 Report

This works shows the evaluation of the influence of IBR on network robustness considering converters. Usueful performance indices are used with the needed reasoning. I have only minor concerns. When using the method DEF (Dissipating Energy Flow), I miss some explanation on this method. Furthermore, I miss some "dt" or similar in the integral terms (5) and (6). In fact one does not know what you are integrating until one arrives to the dicrete equivalent in (7) (if I assume the index "t· to be discrete time, i.e., sampling instant). But in that discrete equivalentIn the index I miss the sampling period for the discretization. It is also hard to understand the use of Delta in "steady-state values" in that method. Please, append some comments on this method and be more clear in the mathematical terms. 

In the References section there is an empty "27." reference. 

Author Response

Point 1: This works shows the evaluation of the influence of IBR on network robustness considering converters. Useful performance indices are used with the needed reasoning. I have only minor concerns. When using the method DEF (Dissipating Energy Flow), I miss some explanation on this method. Furthermore, I miss some "dt" or similar in the integral terms (5) and (6). In fact one does not know what you are integrating until one arrives to the discrete equivalent in (7) (if I assume the index "t· to be discrete time, i.e., sampling instant). But in that discrete equivalent In the index I miss the sampling period for the discretization. It is also hard to understand the use of Delta in "steady-state values" in that method. Please, append some comments on this method and be more clear in the mathematical terms.

In the References section there is an empty "27." reference.

Response 1:

 First of all, we apologize for the error in the equation(7). So, Equation 7 was corrected. In addition, the subscript was corrected to prevent confusion in the equation. The DEF method was cited to prove the results of IILSCR. However, the explanation for the DEF does not seem to have been done properly. We apologize for this. So, we added a description of the equation as shown below. (page 12, line 290-313)

We apologize for empty “27” reference, and we have revised the empty line that is unnecessary.

We have attempted to correct all of the errors throughout the text. We hope the revised manuscript will better meet the requirements of your journal for publication. We thank the editor and the reviewer of the Sustainability once again for the constructive review of our paper.

 Sincerely yours,

Dohyuk Kim, PhD candidate

Department of Engineering

Korea University

Seoul, Korea

Tel : 82-2-3290-3697

E-mail : [email protected]
